

# Land-use and land-cover change carbon emissions between 1901 and 2012 constrained by biomass observations

Wei Li[1], Philippe Ciais[1], Shushi Peng[1,2], Chao Yue[1], Yilong Wang[1], Martin Thurner[3], Sassan S. Saatchi[4], Almut Arneth[5], Valerio Avitabile[6], Nuno Carvalhais[7,8], Anna B. Harper[9], Etsushi Kato[10], Charles Koven[11], Yi Y. Liu[12], Julia E.M.S. Nabel[13], Yude Pan[14], Julia Pongratz[13], Benjamin Poulter[15], Thomas A. M. Pugh[5,16], Maurizio Santoro[17], Stephen Sitch[18], Benjamin D. Stocker[19,20], Nicolas Viovy[1], Andy Wiltshire[21], Rasoul Yousefpour[13,‡], Sönke Zaehle[7]

[1]Laboratoire des Sciences du Climat et de l'Environnement, LSCE/IPSL, CEA-CNRS-UVSQ, Université Paris-Saclay, F-91191 Gif-sur-Yvette, France
[2]Sino-French Institute for Earth System Science, College of Urban and Environmental Sciences, Peking University, Beijing 100871, China
[3]Department of Environmental Science and Analytical Chemistry (ACES) and the Bolin Centre for Climate Research, Stockholm University, SE-106 91 Stockholm, Sweden
[4]Jet Propulsion Laboratory, California Institute of Technology, 4800 Oak Grove Drive, Pasadena, CA 91109, USA
[5]Karlsruhe Institute of Technology, Institute of Meteorology and Climate Research - Atmospheric Environmental Research (IMK-IFU), Garmisch-Partenkirchen, Germany
[6]Centre for Geo-Information and Remote Sensing, Wageningen University & Research, Droevendaalsesteeg 3, 6708PB Wageningen, The Netherlands
[7]Department for Biogeochemical Integration, Max-Planck-Institute for Biogeochemistry, Jena, Germany
[8]CENSE, Departamento de Ciências e Engenharia do Ambiente, Faculdade de Ciências e Tecnologia, Universidade NOVA de Lisboa, Caparica, Portugal
[9]College of Engineering, Mathematics, and Physical Sciences, University of Exeter, Exeter, UK
[10]Institute of Applied Energy, Minato, Tokyo 105-0003, Japan
[11]Climate and Ecosystem Sciences Department, Lawrence Berkeley Lab, Berkeley, CA, USA
[12]ARC Centre of Excellence for Climate Systems Science & Climate Change Research Centre, University of New South Wales, Sydney, New South Wales 2052, Australia
[13]Max Planck Institute for Meteorology, Hamburg, Germany
[14]USDA Forest Service, Durham, New Hampshire, USA
[15]Department of Ecology, Montana State University, Bozeman, MT 59717
[16]School of Geography, Earth & Environmental Science and Birmingham Institute of Forest Research, University of Birmingham, Birmingham, B15 2TT, United Kingdom
[17]GAMMA Remote Sensing, 3073 Gümligen, Switzerland
[18]College of Life and Environmental Sciences, University of Exeter, Exeter, UK
[19]Climate and Environmental Physics, and Oeschger Centre for Climate Change Research, University of Bern, Bern, Switzerland
[20]Imperial College London, Life Science Department, Silwood Park, Ascot, Berkshire SL5 7PY, UK
[21]Met Office Hadley Centre, Exeter, Devon. EX1 3PB, UK
[‡]Current address: Chair of Forestry Economics and Forest Planning, University of Freiburg, 79106 Freiburg, Germany

*Correspondence to*: Wei Li (wei.li@lsce.ipsl.fr)

40



**Abstract.** The use of dynamic global vegetation models (DGVMs) to estimate $CO_2$ emissions from land-use and land-cover change (LULCC) offers a new window to account for spatial and temporal details of emissions, and for ecosystem processes affected by LULCC. One drawback of DGVMs however is their large uncertainty. Here, we propose a new method of using satellite- and inventory-based biomass observations to constrain historical cumulative LULCC emissions ($E_{LUC}^c$) from an

ensemble of nine DGVMs based on emerging relationships between simulated vegetation biomass and $E_{LUC}^c$. This method is applicable at global and regional scale. Compared to the large range of $E_{LUC}^c$ in the original ensemble (94 to 273 Pg C) during 1901-2012, current biomass observations allow us to derive a new best estimate of 155 ±50 (1-σ Gaussian error) Pg C. The constrained LULCC emissions are higher than prior DGVM values in tropical regions, but significantly lower in North America. Our approach of constraining cumulative LULCC emissions based on biomass observations reduces the

uncertainty of the historical carbon budget, and can also be applied to evaluate the impact of land-based mitigation activities.

## 1 Introduction

Carbon emissions from land-use and land-cover change (LULCC) are part of the human perturbation to the global carbon cycle (Houghton et al., 2012; Le Quéré et al., 2015) and started in fact before the Industrial Era when fossil fuel $CO_2$ emissions appeared. Since 1850, estimated cumulative LULCC emissions, $E_{LUC}^c$, represent one-third of total cumulative

anthropogenic $CO_2$ emissions (Boden et al., 2013; Houghton et al., 2012; Le Quéré et al., 2015). Annual LULCC emissions have been higher than those from fossil fuel burning until the 1930s (Boden et al., 2013; Houghton et al., 2012; Le Quéré et al., 2015), and today represent a smaller but persistent perturbation in the global carbon cycle. Unlike fossil fuel emissions, relative uncertainties in LULCC emissions are high, due to the difficulty of assessing this flux from measurements. Some progress has been made to better quantify gross tropical deforestation emissions by combining spatial biomass data with

satellite-derived maps delineating forest cover loss (Harris et al., 2012). But such spatially resolved data are not available beyond the last decade and provide only gross deforestation emissions, i.e. do not track the regrowth of secondary ecosystems or legacy soil carbon losses that can persist long after deforestation.

Bookkeeping models (Hansis et al., 2015; Houghton, 1999) based on historical LULCC area data and tabulated functions of carbon losses and gains are one approach to estimating $E_{LUC}^c$, but they do not include the effects of environmental changes on

carbon stocks before and after LULCC happens (Gasser and Ciais, 2013; Pongratz et al., 2014). The bookkeeping model of Houghton (1999) used for the annual update of the global carbon budget (Le Quéré et al., 2015) is based on regionally aggregated data, and does not consider spatial differences of LULCC fluxes within a region. Alternatively, the estimated LULCC fluxes by dynamic global vegetation models (DGVMs) account for spatial and temporal variations of carbon stock densities and land-cover change, as well as for delayed ("legacy") carbon fluxes. In DGVMs, LULCC fluxes are related to

environmental conditions through simulated carbon cycle processes, i.e., net primary production (NPP) and respiration resulting in changes of biomass and soil carbon stocks are simulated with variable atmospheric $CO_2$ concentration and



climate. Yet, LULCC emissions from DGVMs differ greatly, even when these models are prescribed with the same inputs of land-cover change data (such as time-variable areas of pasture and crops) (Pitman et al., 2009). Several factors are responsible for differences of $E_{LUC}^c$ among DGVMs, including 1) different representations of processes that determine the carbon densities of vegetation and soils subject to land-use change; 2) using dynamic vegetation or prescribing a fixed

vegetation distribution; and 3) use of different rules assigning how natural vegetation types change to agricultural areas (Peng et al., 2016; Pitman et al., 2009; Reick et al., 2013).

Carbon initially stored in forest biomass contributes the predominant portion of the LULCC emissions after deforestation (Hansis et al., 2015). Thus, an accurate representation of the biomass carbon density exposed to LULCC is crucial to reduce uncertainties of DGVM-based $E_{LUC}^c$ estimates. Global biomass datasets based on inventories and satellites recently became

available. These datasets (Table 1) provide the spatially distributed biomass carbon density at regional or global scales (Avitabile et al., 2016; Baccini et al., 2012; Carvalhais et al., 2014; Liu et al., 2015; Pan et al., 2011; Saatchi et al., 2011; Santoro et al., 2015; Thurner et al., 2014), but differ in terms of their coverage of aboveground or belowground biomass and whether they provide only forest biomass or biomass for all vegetation types.

In this study, we propose a new method to combine recent satellite- and inventory-based biomass datasets to constrain $E_{LUC}^c$

simulated by DGVMs (Figure 1). We analyzed the outputs from nine DGVMs (Table 2) of the Trends in Net Land-Atmosphere Exchange (TRENDY-v2) project (Sitch et al., 2015) (http://dgvm.ceh.ac.uk/node/9) and developed global and regional regressions between initial biomass in 1901 and present-day biomass (average of 2000-2012), and regressions between $E_{LUC}^c$ during 1901-2012 and initial biomass across the DGVMs. The former set of regressions is used to extrapolate present-day observation-based biomass (Table 1) to initial biomass in the year 1901. The latter set of regressions is applied

to provide an emerging constraint on $E_{LUC}^c$ as a function of initial biomass (Figure 1). Using the Gaussian uncertainties associated with the observation-based biomass datasets and the uncertainties in the two regressions, uncertainties of $E_{LUC}^c$ can be reduced by providing Gaussian errors after applying the biomass-constraint, compared to the original large range from TRENDY models.

## 2 Materials and Methods

### 2.1 LULCC emissions and biomass from the DGVMs

The DGVMs in TRENDY v2 conducted two simulations (labeled S2 and S3) between 1860 (except JSBACH from 1850, Table 2) and 2012, with outputs quantifying LULCC emissions over the period 1901-2012 (Sitch et al., 2015). Both simulations are performed with changing climate and $CO_2$ concentration, but one (called S3) has variable LULCC maps based on Land Use Harmonization (LUH) dataset (Hurtt et al., 2011) (with an extension until 2012), and the other (called

S2) has a time-invariant land-cover map representing the state in 1860. The difference of net biome production (NBP, the net carbon exchange between the biosphere and the atmosphere) between these two simulations (S3 and S2) defines modeled





LULCC emissions. This definition of LULCC emissions includes the "lost sink capacity" (called "altered sink capacity" in (Gasser and Ciais, 2013) and "the loss of additional sink capacity" in (Pongratz et al., 2014)), because simulated NBP in the S2 simulation without LULCC represents a net sink over areas affected by LULCC in S3. Modeled LULCC emissions include the legacy emissions from soil carbon losses, and emissions from wood and other products produced by LULCC, as

far as the latter are included in the TRENDY v2 models (Table 2). The DGVMs used in this study are CLM4.5 (Oleson et al., 2013), JSBACH (Reick et al., 2013), JULES3.2 (Best et al., 2011; Clark et al., 2011), LPJ (Sitch et al., 2003), LPJ-GUESS (Smith et al., 2001), LPX-Bern (Stocker et al., 2014), ORCHIDEE (Krinner et al., 2005), VISIT (Ito and Inatomi, 2012; Kato et al., 2013), and OCN (Zaehle and Friend, 2010). Each DGVM is described briefly in Table 2.

Identifying the LULCC-affected grid cells in each model is critical, because only biomass in these grid cells should be used

to constrain LULCC emissions. Grid cells affected by LULCC differ among models. Although all models share the same pasture and cropland areas from LUH dataset (Hurtt et al., 2011), the models have different numbers of PFT, use different PFT definitions and have different allocation rules for translating the shared agricultural data into the new vegetation cover (Peng et al., 2016; Pitman et al., 2009; Reick et al., 2013). As a result, there is no unified map to determine the LULCC-affected grid cells in all models. For the same reasons, the forest areas and the LULCC types are also different among

models.

In this study, we adopted the "deforestation grid cells" in their corresponding PFT maps as a criterion to locate the LULCC-affected grid cells from DGVM outputs. Thus we used the PFT maps from each model to first calculate the temporal change of forest area (total area of all forest PFTs) during 1901-2012 and then select the grid cells that experienced deforestation by comparing the forest area maps between year 1901 and year 2012 (net deforestation). This procedure produces a good

approximation, given the continuously decreasing trend of forest area in LULCC hotspot regions like South and Central America (Figure 2). We also tested an alternative method to determine the LULCC-affected grid cells in TRENDY model outputs, i.e., PFT maps were compared year-by-year during 1901-2012, and grid cells with deforestation were selected (gross deforestation). This method tends to give a greater number of LULCC-affected grid cells, reducing the goodness of fitting in the regression between the biomass in 1901 and $E^c_{LUC}$ during 1901-2012 (Figure S1 and Figure S2). Therefore, the

method of gross deforestation is not used for further analyses.

We verified that deforestation grid cells are responsible for most of the total net LULCC flux. In fact, the average of the different model simulations of LULCC emissions from deforestation grid cells between 1901 and 2012 is approximate 90% of the total LULCC emissions from all grid cells (Figure S1). The LULCC emissions in this study are thus taken to equal the sum of LULCC emissions from the selected deforestation grid cells using our criterion. It should be noted that, although only

deforestation is used as a single criterion to define grid cells affected by LULCC in DGVMs, modeled LULCC emissions also include other types of land-use transitions involving pairs of non-forest PFTs in the selected grid cells.

In each model, only biomass in deforestation grid cells is considered. Biomass in the year 1901 is thereby defined as *initial* biomass, and biomass averaged during 2000-2012 is defined as *present* biomass. An ordinary least squares linear regression





is performed with the outputs of all models between initial biomass and $E_{LUC}^c$ from 1901 to 2012, and between the initial and the present biomass at both global and regional scales. Our division of nine regions in the world (Figure 2) for estimating LULCC fluxes is the same as Houghton et al. (1999).

## 2.2 Observation-based biomass datasets

Several biomass datasets (Avitabile et al., 2016; Baccini et al., 2012; Carvalhais et al., 2014; Liu et al., 2015; Pan et al., 2011; Saatchi et al., 2011; Santoro et al., 2015; Thurner et al., 2014) based on inventories and remote sensing can be potentially used to constrain $E_{LUC}^c$ through the set of regressions from DGVMs. However, these biomass datasets cover different parts of biomass (aboveground, belowground or total) and different regions (tropics, Northern Hemisphere or the globe) at different spatial resolutions (Table 1). We choose the global grid-based biomass dataset from Carvalhais et al.

(2014) to derive an observational constraint which brings a best estimate of $E_{LUC}^c$. This map merges the Northern Hemisphere biomass dataset from Thurner et al. (2014) and the tropical biomass dataset from Saatchi et al. (2011). An advantage of this map is its consistency in biomass terms with the outputs of TRENDY models, because it documents aboveground + belowground, and forest + herbaceous biomass (Tables 1, 2). Three other biomass maps are used as alternative datasets for sensitivity tests: 1) the global biomass map from the GEOCARBON project, a merged product of the biomass datasets in the

Northern Hemisphere (Santoro et al., 2015) and tropics (Avitabile et al., 2016); 2) regional biomass estimates from Pan et al. (2011) based on forest inventory data; and 3) the biomass map from Liu et al. (2015) derived from satellite vegetation optical depth. The GEOCARBON (Avitabile et al., 2016; Santoro et al., 2015) and Liu et al. (2015) datasets that only provide aboveground biomass were extended to total forest biomass using the conversion factors for the nine regions (Liu et al., 2015). The global biomass maps from GEOCARBON (Avitabile et al., 2016; Santoro et al., 2015) and Pan et al. (2011) are

only for forest (Table 1), and we do not add the herbaceous biomass to these two datasets because the global herbaceous biomass only accounts for about 3% of the global total biomass (Carvalhais et al., 2014). Note that the uncertainties in the corresponding constrained results using these three alternative datasets do not include the uncertainties of converting aboveground biomass to the total of aboveground and belowground biomass for datasets from  Liu et al. (2015) and GEOCARBON (Avitabile et al., 2016; Santoro et al., 2015) and the uncertainties of ignoring non-woody biomass in datasets

from GEOCARBON (Avitabile et al., 2016; Santoro et al., 2015) and Pan et al. (2011). The biomass maps of Carvalhais et al. (2014), GEOCARBON (Avitabile et al., 2016; Santoro et al., 2015) and Liu et al . (2015) with different spatial resolutions were aggregated to 1°×1° resolution before selecting the deforestation grid cells.

## 2.3 Methods to identify grid cells subject to past deforestation in biomass datasets

It is not practical to use PFT maps from DGVMs to define deforestation grid cells in the observation-based biomass datasets

because PFT maps and forest area change since 1901 differ across DGVMs. Instead, we diagnosed deforestation grid cells in the biomass maps using three harmonized methods (Method-A, Method-B and Method-C). All the methods are based on the




reconstructed historical agricultural area from the History Database of the Global Environment(Klein Goldewijk et al., 2011) (HYDE v3.1) but with different hypotheses regarding how agricultural expansion has affected forests. These harmonized methods are representative of the different rules of assigning LULCC data to natural vegetation types in DGVMs. Method-A assumes that the increase of cropland area in a grid cell between 1901 and 2012 is taken from forest; Method-B assumes that

the increase of cropland and pasture is taken proportionally from all natural vegetation types; and Method-C (like the "BM3" scenario in (Peng et al., 2016)) assumes that the increase of cropland and pasture is first taken from forest and then from natural grassland if no more forest area is available, and that the regional forest area change is set to match the historical forest reconstruction from (Houghton, 2003). Because the biomass distribution in Pan et al. (2011) is given as regional mean values and not resolved on grid cell basis, it is impossible to select deforestation grid cells directly from this dataset using the

above methods. Therefore, for each region, we calculated the ratios of biomass in deforestation grid cells according to Method-A, Method-B and Method-C to the total biomass in all grid cells in each of the other three biomass datasets (Carvalhais et al. (2014), GEOCARBON (Avitabile et al., 2016; Santoro et al., 2015) and Liu et al. (2015)). For each method (Method-A, -B, and -C), the three ratios corresponding to the three biomass datasets were further averaged in each region. The total biomass amount from Pan et al. (2011) in each region was multiplied by the average ratio to derive the biomass

equivalent to using Method-A, Method-B and Method-C for the dataset from Pan et al. (2011).

These three methods applied to the above-listed biomass datasets are also applied as sensitivity tests to select the deforestation grid cells since 1901 in the TRENDY model outputs. Identically, regressions are performed using initial biomass amount and $E_{LUC}^c$ from these selected grid cells. Due to the inconsistencies between the three methods and the historical PFT maps of each DGVM, the biomass amount in 1901 in the selected grid cells using these three methods is

higher than using PFT maps, but the $E_{LUC}^c$ are lower, reflecting a lower representativeness of the deforestation grid cells using these three methods for DGVM outputs (Figure S1). As a consequence, a weaker goodness of regression fit was found between $E_{LUC}^c$ and initial biomass (Figure S2).

## 2.4. Uncertainties of constrained LULCC emissions

The biomass from Method-A, Method-B and Method-C obtained from each dataset is extrapolated into biomass for the year

1901 using the regression between initial biomass and present biomass modeled by the DGVMs. This biomass in 1901 is then applied in the regression between modeled $E_{LUC}^c$ and modeled initial biomass among different DGVMs to calculate constrained $E_{LUC}^c$. In this emerging constraint approach (Figure 1), the uncertainties of constrained $E_{LUC}^c$ are a function of the uncertainties of the observed biomass datasets, of the linear regression goodness of fit for the two regressions (regressions between $E_{LUC}^c$ and the initial biomass, and between the initial and present biomass), and of the slopes of the regressions. The

uncertainty of constrained LULCC emissions is calculated as (Stegehuis et al., 2013):

$$\sigma_{LULCC} = \sqrt{\alpha^2 \sigma_{initial\_biomass}^2 + \sigma_{res\_LULCC}^2} \quad (1)$$



$$\sigma_{initial\_biomass} = \sqrt{\beta^2 \sigma^2_{present\_biomass} + \sigma^2_{res\_biomass}} \quad (2)$$

where $\sigma_{LULCC}$, $\sigma_{initial\_biomass}$ and $\sigma_{present\_biomass}$ are the uncertainties of constrained $E^c_{LUC}$, the uncertainty of initial biomass and the uncertainty of present biomass. $\alpha$ and $\sigma_{res\_LULCC}$ represent the slope and the standard deviation of the residuals from the linear regression fit between $E^c_{LUC}$ and initial biomass. $\beta$ and $\sigma_{res\_biomass}$ represent the slope and standard deviation of the residuals from the linear regression between initial biomass and present biomass.

## 3. Results

### 3.1. Forest area change and cumulative LULCC emissions in DGVMs

As expected, a general decrease of forest area is found between 1901 and 2012, especially in regions subject to extensive deforestation over the last decades, namely South and Central America, south and southeast Asia, and tropical Africa (Figure 2), which is in support of our methods of defining deforestation grid cells, although the forest area in some regions differs substantially across DGVMs. Differences in forest area are large in tropical Africa, North America and the former Soviet Union while they are smaller in South and Central America, and south and southeast Asia (Figure 2). There are several reasons for these differences of forest area: 1) models have different initial distribution of PFTs (the TRENDY v2 protocol only prescribed the same initial area of natural vegetation, but did not specify the PFTs that compose natural vegetation); 2) some models consider only net LULCC, but others have gross LULCC including some sub-grid transitions (Table 2, e.g., see a comparison using the JSBACH model (Wilkenskjeld et al., 2014)); 3) models have different treatments for changing pasture areas (either proportional from natural vegetation or preferential from natural grasslands). In North America, China region and western Europe, the forest area decreased in the first half of the 20$^{th}$ century and then increased in recent decades. Yet, the magnitude of the increase is smaller than that of the previous decrease in these regions, and the global average is net forest loss between 1901 and 2012 (ranging from 2.3 Mkm$^2$ to 16.8 Mkm$^2$ across the nine models).

$E^c_{LUC}$ from the nine DGVMs between 1901 and 2012 ranges from 1.7 (-0.6 to 6.0) Pg C (median and range, a positive number indicating a net cumulative flux to the atmosphere) in north Africa and Middle East to 42.6 (33.5 to 81.4) Pg C in South and Central America, resulting in a global total of 148 (94 to 273) Pg C (Table 3). Tropical Africa and south and southeast Asia have the second large $E^c_{LUC}$ of 21.8 (15.8 to 57.8) and 21.8 (9.6 to 46.6) Pg C, respectively. Although afforestation / reforestation occurred in North America after around 1960, and in China after 2000 (Figure 2), $E^c_{LUC}$ of these two regions are positive since 1901, with median values of 19.9 and 10.7 Pg C, respectively (Table 3).

### 3.2. Relationship between cumulative LULCC emissions and initial biomass

We found a positive linear relationship between $E^c_{LUC}$ and initial biomass in deforestation grid cells of each model, both at global scale and in the regions considered (Figure 3). The coefficients of determination (r$^2$) are 0.61, 0.58 and 0.76 in South




and Central America, south and southeast Asia and tropical Africa, respectively. Due to stable or slightly increasing forest area (Figure 2), the correlation between initial biomass and $E_{LUC}^c$ is small in western Europe (Figure 3). The slopes of the relationships between $E_{LUC}^c$ and initial biomass shown in Figure 3 range from 0.13 Pg C/Pg C in western Europe to 0.63 Pg C/Pg C in north Africa and Middle-East. In tropical regions with intensive LULCC, the slope is similar between south and

southeast Asia (0.36 Pg C/Pg C) and tropical Africa (0.37 Pg C/Pg C), but lower in South and Central America (0.21 Pg C/Pg C). These slopes reflect the sensitivity of cumulative carbon loss to initial biomass carbon stock. They are mainly influenced by the fraction of deforested area relative to the initial forest area in each region, which explains 46% of the variations of the slopes across regions (Figure S3). Differences in biomass density across regions and in the use of gross or net transitions among DGVMs (Table 2) also contribute to variations of slopes.

**3.3. Cumulative LULCC emissions constrained by present-day biomass observations**

There is also a strong positive relationship between initial biomass in 1901 and present-day biomass in grid cells experienced deforestation (Figure 4). The $r^2$ of this regression is higher than 0.92 in most regions, except in North America and China region (0.89 and 0.76, respectively). The regression between present-day and initial biomass was applied to extrapolate current observation-based biomass back to the year 1901. The extrapolated biomass in 1901 is higher than that in the present

day, mainly due to a larger forest area — although it is difficult to discriminate other effects such as $CO_2$ fertilization that might have increased biomass between 1901 and 2012.

Using the chain of emerging constraints between present-day and initial biomass (Figure 4), and between $E_{LUC}^c$ and initial biomass (Figure 3), with all uncertainties being propagated (Eqs. (1) and (2)), we were able to constrain $E_{LUC}^c$ during 1901-2012 by biomass observations (Figure 3, S4, S5, and Table 3). $E_{LUC}^c$ constrained by the biomass dataset of Carvalhais et al.

(2014) is 155 ±50 (mean and 1-σ Gaussian error) Pg C and this estimate is robust to the choice of the methods to define deforestation grid cells in biomass datasets (constrained $E_{LUC}^c$ = 152 ±49, 154 ±50 and 159 ±51 Pg C for Method-A, Method-B and Method-C, respectively). The uncertainties reported in our constrained estimate of $E_{LUC}^c$ include uncertainties in the biomass observations, and in the scatter of the two regressions (Figure 3, 4) used to construct the emerging constraint. Compared to the original global total $E_{LUC}^c$ range (94 to 273 Pg C) in DGVMs, the constrained estimate has a smaller

uncertainty and a larger median value. It should be noted that we summed the biomass uncertainty in each deforestation grid cell to give the regional biomass uncertainty, which gives a maximum uncertainty with a potential assumption that the uncertainties in all grid cells are fully correlated. In reality, the regional biomass uncertainty should be lower, thus leading to a lower uncertainty of constrained $E_{LUC}^c$. However, it is difficult to estimate the error correlations of observation-based biomass between different grid cells at this stage.

Although the constrained global $E_{LUC}^c$ value is only 7 Pg C higher than the median of the original DGVM ensemble (Table 3), larger differences can be found at regional scale (Figure 5). Constrained $E_{LUC}^c$ estimates are higher than the original modeled values in south and southeast Asia, tropical Africa and South and Central America (Table 3). For example, the constrained E





$^c_{LUC}$ is 37.2 ±14.4 Pg C in south and southeast Asia, compared to the original TRENDY median value of 21.8 Pg C (range of 9.6 to 46.6 Pg C) for that region. The constrained emissions are also higher in China region and the Pacific developed region compared to the prior median value (see Table 3). A significantly large reduction in $E^c_{LUC}$ through the emerging constraint is found in North America because of the lower biomass amount from observation-based datasets than from DGVMs. The

original median $E^c_{LUC}$ of that region is 19.9 Pg C (range of 8.6 to 40.8 Pg C) while the constrained result is 10.8 ±7.1 Pg C. Constrained $E^c_{LUC}$ are also lower than original estimates in western Europe, and north Africa & Middle East, although their contributions to the global total emissions are very small (Table 3).

Alternative estimates of $E^c_{LUC}$ constrained by three other biomass datasets (Liu et al. (2015), GEOCARBON (Avitabile et al., 2016; Santoro et al., 2015), and Pan et al. (2011)) are provided in Figure 6 and Table 3. In general, constrained $E^c_{LUC}$ using

biomass maps from Liu et al. (2015) and GEOCARBON (Avitabile et al., 2016; Santoro et al., 2015) are slightly higher (7 Pg C on average) than those from Carvalhais et al. (2014). The biomass dataset from Pan et al. (2011) leads to lower LULCC emission estimates at global scale, mainly due to a lower estimate in south and southeast Asia (Table 3) compared to the other products. In the Pacific developed region, GEOCARBON-based estimates (Avitabile et al., 2016; Santoro et al., 2015) are much higher than those from Carvalhais et al. (2014), because the latter has a gap in their biomass map in the southern

part of Australia (Carvalhais et al., 2014).

## 4. Discussion

Our approach to constraining $E^c_{LUC}$ from an ensemble of DGVMs provides a best estimate that is between those from two bookkeeping models (~130 Pg C from Houghton et al. (2012) and 212 Pg C for the default dataset from Hansis et al. (2015)). Although the bookkeeping model from Hansis et al. (2015) was driven by the same agricultural land use maps as the

TRENDY models (the model of Houghton et al. (2012) uses FRA/FAO data), $E^c_{LUC}$ from Hansis et al. (2015) is different from that constrained from the DGVMs. Differences of estimates between DGVMs and bookkeeping models have been attributed to different definitions of LULCC emissions (Pongratz et al., 2014; Stocker and Joos, 2015). Indeed, LULCC emissions from DGVM simulations in TRENDY include the "missed sink capacity in the deforested area" (Gasser and Ciais, 2013; Pongratz et al., 2014), and so, all else being equal, should simulate higher emissions than bookkeeping models which do not include

this term. However, bookkeeping models take forest degradation into account, while this process is ignored in DGVMs. Bookkeeping models also represent shifting cultivation (resulting in larger sub-grid scale gross land transitions as opposed to net transitions) and wood harvest; processes which are accounted for in only a subset of the TRENDY models (see Table 2). In addition to different driving LULCC area data, differences between the two bookkeeping models were discussed by Hansis et al. (2015); for example, Houghton et al. (2012) assumed a preferential allocation of pastures on natural grasslands,

while Hansis et al. (2015) assumed a proportional allocation of both cropland and pasture on all available natural vegetation types.



We are aware that our truncated diagnostic of a set of deforestation grid cells, instead of grid cells affected by all LULCC types, is an under-estimate of the total area subject to LULCC, because we ignore grid cells that experienced land-use transitions between non-forest vegetation only (e.g. only conversions from grasslands to cropland happening in a grid cell). However, the conversion of forest to croplands and pasture dominates the total net LULCC flux (Houghton, 2003, 2010),

while the contribution of transitions between non-forest vegetation and agriculture to $E_{LUC}^c$ is comparatively small (Figure S1). In fact, the annual LULCC emission from deforestation was estimated to be 2.2 Pg C yr$^{-1}$ during 1990s, and the total emissions from other activities (e.g., afforestation, reforestation, non-forest transitions) is nearly neutral (Houghton, 2003).

The lack of direct biomass observations at the initial state forces us to hindcast biomass in 1901 based on present-day observations; an extrapolation that also comes with uncertainties. Some of the observed biomass datasets only cover forests,

and satellite measurements usually quantify aboveground biomass carbon stocks and not total biomass stocks (Table 1). In addition, the regression of modeled biomass between 1901 and 2000-2012 (average) to extrapolate the biomass amount in 1901 is only a statistical approach. This regression cannot be mechanistically explained because its slope and intercept are impacted by multiple factors in the models like land clearing, secondary vegetation regrowth, the $CO_2$ fertilization, climate, disturbances, and the nutrient limitation on biomass. Despite these uncertainties, the high coefficient of determination in the

regression increases our confidence in the biomass extrapolation to 1901. Methods of defining deforestation grid cells (Method-A, Method-B and Method-C) to be associated with a certain biomass dataset had very small influence on our results (Table 3).

The required model outputs for carbon stocks and fluxes in the TRENDY project are not PFT-specific; only the mean PFT-mixed variables in each grid cell are required. Such an aggregation prevents a rigorous separation of biomass between forest

and other biomes in each grid cell. It was thus impossible for us to calculate individual contributions of different LULCC types to the overall LULCC emissions, which induces uncertainties when matching model results with observed forest biomass distributions (e.g. only forest biomass in datasets from GEOCARBON (Avitabile et al., 2016; Santoro et al., 2015) and Pan et al. (2011)). Therefore, we suggest that the next generation of DGVM comparisons reports PFT-specific carbon stock and fluxes, and other model inter-comparison exercises should follow suit. The approach of using multiple biomass

observation datasets to constrain the LULCC emissions could also be applied in other modeling projects, such as Coupled Model Intercomparison Project Phase 5 (CMIP5) and CMIP6.

Currently, the uncertainties in the satellite-based biomass datasets are relatively large at the pixel level (< 1 km) that may introduce uncertainties of constrained cumulative LULCC emissions, depending on the forest types and biomass range. For example, on the average at the global scale, the uncertainty at the modeling grid cells (0.5° × 0.5°) is about one-third of the

mean biomass (Carvalhais et al., 2014) and relatively smaller for high biomass areas in tropics (Avitabile et al., 2016; Saatchi et al., 2011).

The main sources of uncertainties in satellite-based biomass datasets depend on the specific product, spatial resolution of the datasets, and the methodology used to validate the data. For instance, in the case of radar remote sensing used for biomass



mapping in Northern Hemisphere boreal and temperate forests, the uncertainty is largely due to the sensitivity of the signal to other properties than vegetation structure (e.g. moisture), the influence of non-forest vegetation on the signal (especially in fragmented landscapes; (Santoro et al., 2015)), and uncertainties in additional datasets (allometric databases, land cover) used for conversion of satellite measurements to biomass estimates (Thurner et al., 2014). At the pixel level and modeling

grid cells, uncertainties may also be strongly influenced by the quality and size of inventory data used for validation and the significant mismatch between pixel area and the plot data, and the difference of between the dates of satellite and ground observations (Saatchi et al., 2015, 2011; Thurner et al., 2014).

Moreover, the satellite-derived biomass datasets used in this study represent different dates. The tropical biomass products represent circa 2000 status of forests, whereas the boreal and temperate biomass maps are based on spaceborne radar data

from the year 2010. These differences in the date of observations introduce additional uncertainty in the biomass estimates due to changes of forest cover from disturbance and recovery and land use activities (Hurtt et al., 2011) occurring annually and regionally.

However, in boreal/temperate as well as in tropical regions, the estimated relative uncertainties were lowest in high biomass areas (Avitabile et al., 2016; Thurner et al., 2014), which dominates the contribution to our results. Moreover, the relatively

high accuracy of biomass datasets when aggregated to modeling grid cells from higher resolution maps (< 1 km) (Saatchi et al., 2011; Thurner et al., 2014) suggest that biomass datasets implemented in our study provide a realistic representation of carbon stocks to constrain the historical cumulative LULCC emissions from vegetation.

## 5. Conclusions

Uncertainties in LULCC carbon emissions are relatively large, compared to other terms in the global carbon budget. The

wide spread is partly due to the differences in model structure but also because of the difficulty in constraining models by observations of LULCC, particularly emissions resulting from deforestation. We use satellite and inventory-based biomass observations to constrain simulated historical LULCC emissions based on relationships between simulated vegetation biomass and cumulative LULCC emissions in nine DGVMs. We propose an observationally constrained global cumulative LULCC emission of 155 ±50 Pg C during 1901 and 2012. Although the uncertainties in current observation-based biomass

datasets are relatively high, as more accessible and accurate observation data become available, many data-driven opportunities are being created to improve the accuracy of DGVM predictions.

### Acknowledgements

W.L., C.Y., T.A.M.P. and A.A. were supported by the European Commission-funded project LUC4C (grant No. 603542). P.C. and S.P. acknowledge support from the European Research Council through Synergy grant ERC-2013-SyG-610028

"IMBALANCE-P". J.P., J.N., and R.Y. were supported by the German Research Foundation's Emmy Noether Program (PO





1751/1-1). B.D.S. was supported by the Swiss National Science Foundation and FP7 funding through project EMBRACE (282672). A.H. was supported by the UK Natural Environment Research Council Joint Weather and Climate Research Programme. M.T. acknowledges funding from the Vetenskapsrådet grant 621-2014-4266 of the Swedish Research Council. The biomass maps and model outputs can be freely accessed following instructions in the original publications. All the biomass constrained LULCC emission data are freely obtained from WL (email: wei.li@lsce.ipsl.fr).

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



## 1.2 Subsection (as Heading 2)

Ut rutrum, sapien et vulputate molestie, augue velit consectetur lectus, bibendum porta justo odio lobortis ligula. In in urna nec arcu iaculis accumsan nec et quam. Integer ut orci mollis, varius justo vitae, pellentesque leo. Vestibulum eu finibus nisl. 5   Cras ac arcu urna. Duis ut pellentesque urna.

### 1.2.1 Subsubsection (as Heading 3)

In placerat dictum urna ut interdum. Etiam vel nibh vulputate, scelerisque purus in, congue eros. Pellentesque at nisi at nunc sagittis cursus. Mauris euismod tellus at mi tempor, sit amet finibus ante tincidunt. Aenean id ornare neque. Cras ut sapien quis erat pretium ultricies. Integer vulputate ante nec elementum tristique. Ut. Lorem ipsum dolor sit amet, consectetur 10   adipiscing elit. Mauris dictum, nibh ut condimentum pharetra, quam ligula varius est, sed vehicula massa erat ut metus. In eget metus lorem. Fusce vitae ante dictum, elementum sem non, lacinia dui. Integer tellus tortor, convallis et aliquam non, dictum vel mauris. Quisque maximus mollis dui, a mollis mauris vehicula in. Duis dui ligula, suscipit ac lectus vitae, fringilla euismod diam. Suspendisse a elit ut leo pharetra cursus sed quis diam. Nullam dapibus, ante vitae congue egestas, sem ex semper orci, vel sodales sapien nibh sed lectus. Etiam vehicula lectus quis orci ultricies dapibus. In sit amet lorem 15   egestas, pretium sem sed, tempus lorem. Quisque cursus massa sed urna congue, ac convallis neque consectetur. Proin faucibus neque non metus mollis, suscipit pretium nisl blandit. In hac habitasse platea dictumst. Nam laoreet augue eu odio eleifend, non posuere quam pulvinar. Integer sit amet leo vitae nisl facilisis tristique calculated following Eq. (1):

$$Y = \frac{\Delta M_0}{\Delta[\text{isoprene}]} \, , \qquad\qquad (1)$$

where $\Delta M_0$ is Ut rutrum, sapien et vulputate molestie, augue velit consectetur lectus, bibendum porta justo odio lobortis 20   ligula. In in urna nec arcu iaculis accumsan nec et quam. Integer ut orci mollis, varius justo vitae, pellentesque leo. Vestibulum eu finibus nisl. Cras ac arcu urna. Duis ut pellentesque urna. In placerat dictum urna ut interdum. Etiam vel nibh vulputate, scelerisque purus in, congue eros. Pellentesque at nisi at nunc sagittis cursus. Mauris euismod tellus at mi tempor, sit amet finibus ante tincidunt. Aenean id ornare neque. Cras ut sapien quis erat pretium ultricies. Integer vulputate ante nec elementum tristique. Ut.



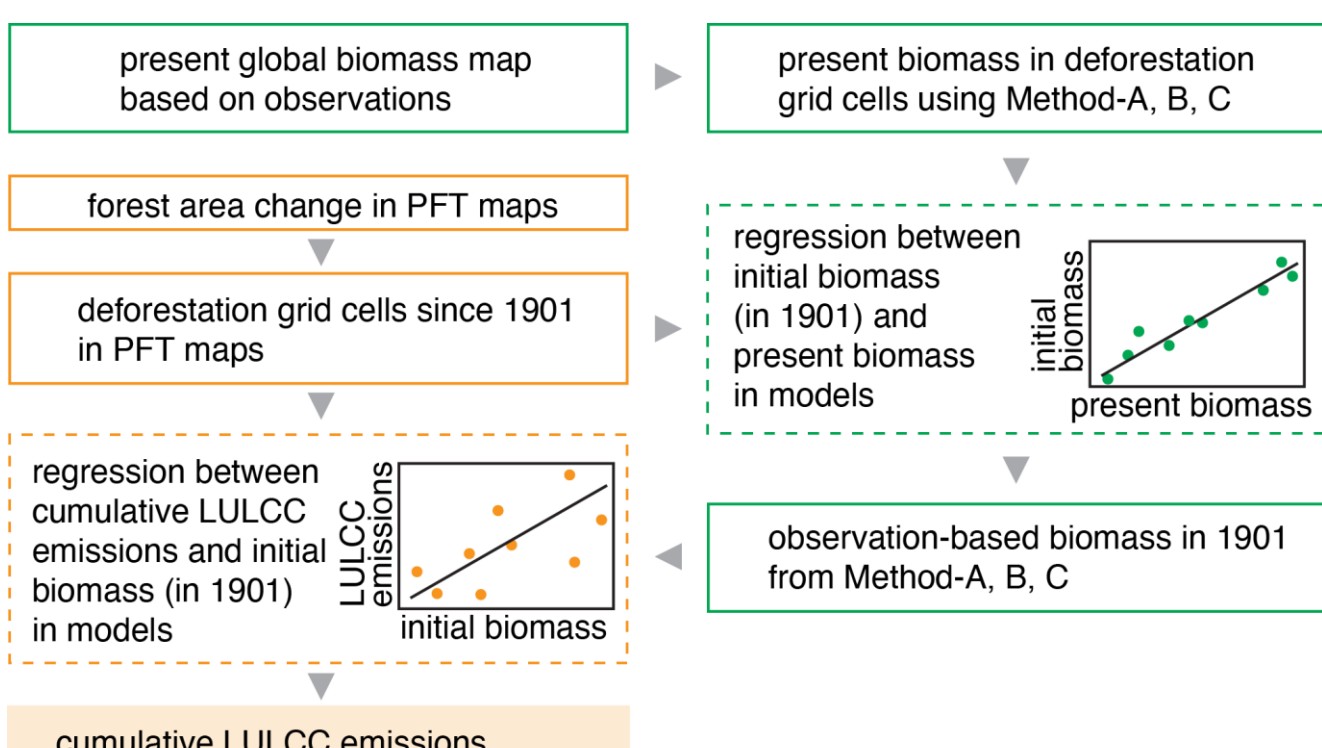

**Figure 1.** The framework of this study.

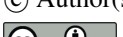



Figure 2. Temporal change of forest area from TRENDY v2 models in each of the nine regions. Differences between models arise from their specific vegetation maps and rules through which natural PFTs are chosen to give land to agriculture.




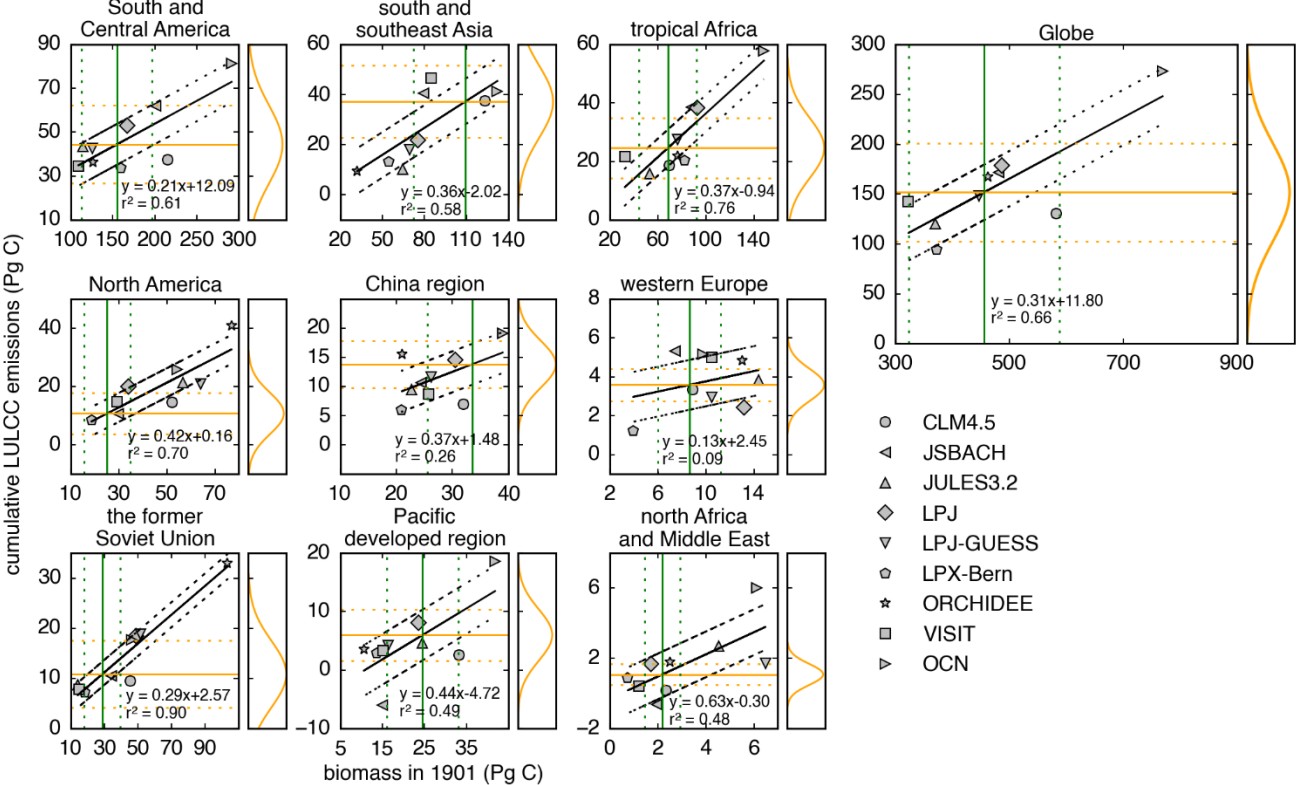

**Figure 3. Relationship between biomass in 1901 and cumulative land-use and land-cover change (LULCC) emissions during 1901-2012 across the nine TRENDY v2 models. The black solid line is the linear regression line. The vertical green solid line indicates the reconstructed biomass in 1901 from Carvalhais et al. (2014) by applying Method-A (all the increase of cropland in HYDE v3.1 data from forest, see Figs. S4 and S5 for results of Method-B and Method-C) to define deforestation grid cells. The orange solid horizontal line indicates the cumulative LULCC emissions constrained by reconstructed biomass in 1901. Dashed lines represent 1-σ uncertainties. The probability density function of the constrained cumulative LULCC emissions is shown on the right.**





**Figure 4. The relationship between initial biomass in 1901 and present biomass (average of biomass from 2000 to 2012) across the TRENDY v2 models for each region. Dashed line is the 1:1 line.**

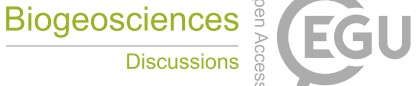

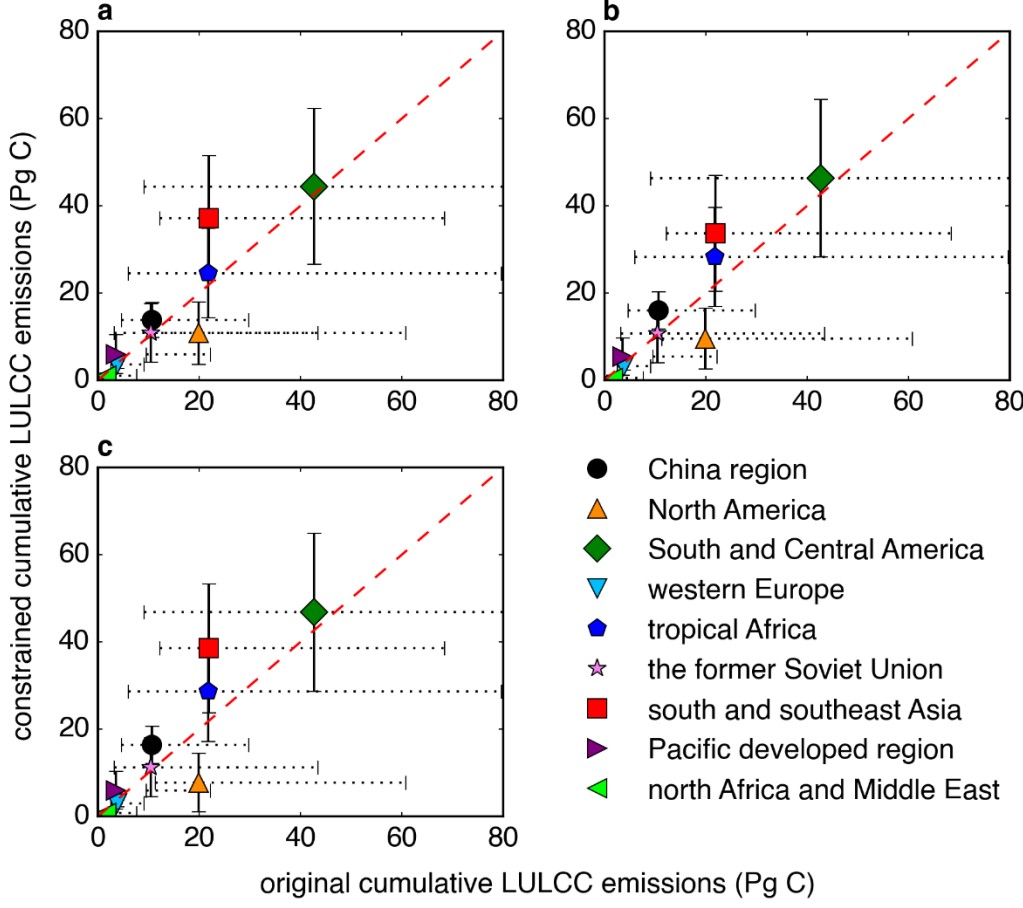

**Figure 5.** Comparisons between the original TRENDY land-use and land-cover change (LULCC) emissions and the cumulative LULCC emissions constrained by biomass dataset from Carvalhais et al. (2014). (a), (b) and (c) are results from Method-A, Method-B, and Method-C, respectively. Solid and dotted lines represent the 1-σ Gaussian error after biomass constraint and the original modeled range. Dashed line is the 1:1 line.



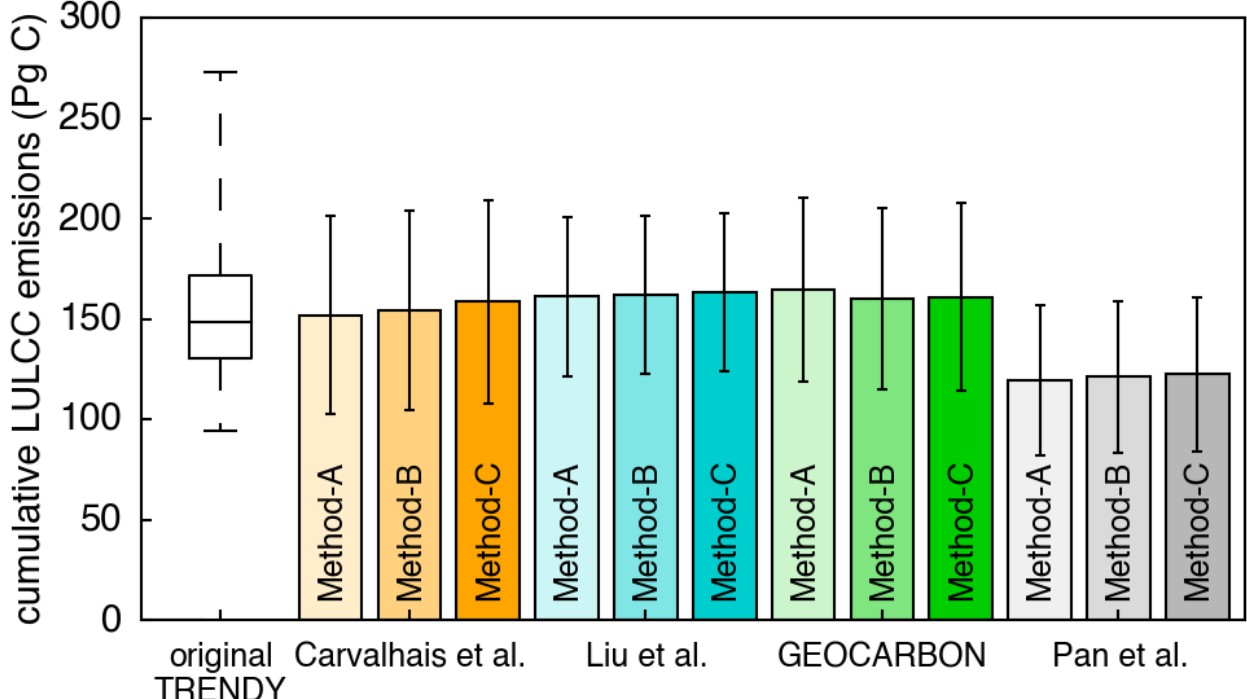

**Figure 6. The global cumulative land-use and land-cover change (LULCC) emissions during 1901-2012 from original TRENDY models and from the estimates constrained by different biomass datasets with different methods to define deforestation grid cells. The whisker-box plot represents the minimum and maximum values, 25th and 75th percentiles and the median value of original TRENDY models. The error bars in the bar plot represent the 1-σ Gaussian errors.**



**Table 1.** The different biomass datasets based on observations. The information of biomass from TRENDY v2 project is also listed for comparison.

| dataset | coverage | resolution | biome type | aboveground / belowground | note |
|---|---|---|---|---|---|
| Thurner et al. (Thurner et al., 2014) | 30°N~80°N | 0.01 degree | forest | aboveground + belowground | Growing stock volume from Santoro et al. (Santoro et al., 2015) |
| Saatchi et al. (Saatchi et al., 2011) | 30°N – 40°S | 1 km | forest | aboveground | |
| Carvalhais et al. (Carvalhais et al., 2014) | global (without South Australia) | 0.5 degree | forest herbaceous | + aboveground + belowground | a merged map of Thurner et al. (Thurner et al., 2014) and Saatchi et al. (Saatchi et al., 2011) |
| Baccini et al. (Baccini et al., 2012) | 23°N – 23°S | 500 m | forest | aboveground | |
| Liu et al. (Liu et al., 2015) | global | 0.25 degree | all | aboveground | calibration based on Saatchi et al. (Saatchi et al., 2011) |
| Avitabile et al. (Avitabile et al., 2016) | 30°N – 40°S | 1 km | forest | aboveground | a fusion of Saatchi et al. (Saatchi et al., 2011) and Baccini et al. (Baccini et al., 2012) |
| Santoro et al. (Santoro et al., 2015) | 30°N+ | 0.01 degree | forest | aboveground | sharing growing stock volume with Thurner et al. (Thurner et al., 2014) |
| Pan et al. (Pan et al., 2011) | global | regional | forest | aboveground + belowground | based on FAO data |
| TRENDY v2 | global | 1 degree | all | aboveground + belowground | |



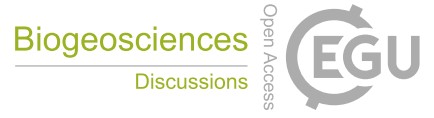

**Table 2.** Description of TRENDY model set-ups used in this study.

| model | PFT number | allocation rules of changes in agriculture area | spatial resolution | dynamic vegetation activated | wood harvest | shifting agriculture | explicit gross LULCC transitions | start of simulation | transient reference |
|---|---|---|---|---|---|---|---|---|---|
| CLM4.5 | 17 | pasture; crop model exists but not used in these simulations | 0.9°×1.25° | no | yes | no | no | 1860 | (Oleson et al., 2013) |
| JSBACH | 12 | proportional allocation of cropland; preferential allocation of pasture on natural grassland | T63[a] | no | yes | yes | yes | 1850 | (Reick et al., 2013) |
| JULES3.2 | 5 | crop and pasture added together to create a single agricultural mask, where trees and shrubs are excluded from growing. There is no assumption for which PFTs the agriculture replaces | N96[b] | yes | yes | no | no | 1860 | (Clark et al., 2011; Best et al., 2011) |
| LPJ | 9 | crop and pasture were added together to create a single managed lands fraction | 0.5°×0.5° | yes | no | no | no | 1860 | (Sitch et al., 2003) |
| LPJ-GUESS | 11 | proportional allocation of cropland and pasture | 0.5°×0.5° | yes | no | no | no | 1860 | (Smith et al., 2001) |
| LPX-Bern | 9 | proportional allocation of cropland and pasture | 1.0°×1.0° | yes | no | no | no | 1860 | (Stocker et al., 2014) |
| ORCHIDEE | 13 | proportional allocation of cropland and pasture | 2°×2° | no | no | no | no | ?? | (Krinner et al., 2005) |
| VISIT | 16 | no specific rule applied because only natural PFT exists for primary and secondary land in a grid cell | 0.5°×0.5° | no | yes | yes | yes | 1860 | (Kato et al., 2013; Ito and Inatomi, 2012) |
| OCN | 12 | proportional allocation of cropland and pasture | 2.5°×3.75° | xno | no | no | no | 1860 | (Zaehle and Friend, 2010) |

[a]T63 grid has an approximate resolution of 1.9°×1.9°.

[b]N96 resolution is equivalent to 1.25°latitude × 1.875°longitude.

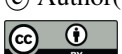

**Table 3.** The global and regional cumulative land-use and land-cover change (LULCC) emissions (Pg C) during 1901-2012 from original TRENDY models and from the estimates constrained by different biomass datasets with different methods to define deforestation grid cells.

| | TRENDY | | | Carvalhais et al. (Carvalhais et al., 2014) | | | Liu et al. (Liu et al., 2015) | | | GEOCARBON (Avitabile et al., 2016; Santoro et al., 2015) | | | Pan et al. (Pan et al., 2011) | | |
|---|---|---|---|---|---|---|---|---|---|---|---|---|---|---|---|
| | median | min | max | Method-A | Method-B | Method-C | Method-A | Method-B | Method-C | Method-A | Method-B | Method-C | Method-A | Method-B | Method-C |
| China region | 10.7 | 6.0 | 19.1 | 13.8±4.0 | 16.0±4.3 | 16.3±4.3 | 10.5±3.1 | 11.1±3.1 | 11.1±3.1 | 10.0±4.0 | 10.5±4.5 | 10.5±4.6 | 7.3±2.9 | 7.6±2.9 | 7.7±2.9 |
| North America | 19.9 | 8.6 | 40.8 | 10.8±7.1 | 9.6±7.0 | 7.8±6.7 | 14.7±6.9 | 13.6±6.8 | 9.3±6.6 | 17.8±8.3 | 15.4±7.7 | 13.0±7.6 | 9.5±6.4 | 8.5±6.4 | 6.7±6.4 |
| South and Central America | 42.6 | 33.5 | 81.4 | 44.4±17.8 | 46.4±18.1 | 46.8±18.2 | 48.3±17.0 | 50.1±17.0 | 50.6±17.0 | 44.5±16.6 | 46.4±16.7 | 46.8±16.8 | 43.1±17.0 | 44.8±17.1 | 45.1±17.2 |
| western Europe | 3.8 | 1.2 | 5.3 | 3.6±0.8 | 3.2±0.8 | 3.0±0.8 | 4.1±0.8 | 3.4±0.8 | 3.2±0.8 | 5.0±1.2 | 3.8±0.9 | 3.4±0.8 | 3.6±0.8 | 3.2±0.8 | 3.0±0.8 |
| tropical Africa | 21.8 | 15.8 | 57.8 | 24.6±10.3 | 28.2±11.4 | 28.6±11.5 | 31.4±8.8 | 36.3±9.4 | 36.9±9.4 | 22.7±12.8 | 26.2±4.6 | 26.4±16.1 | 23.8±10.3 | 27.5±11.4 | 27.8±11.5 |
| the former Soviet Union | 10.5 | 7.2 | 33.0 | 10.9±6.7 | 10.7±6.7 | 11.3±6.8 | 14.2±6.5 | 14.3±6.5 | 14.9±6.5 | 14.7±6.6 | 13.0±6.5 | 13.5±6.5 | 10.1±6.2 | 9.8±6.2 | 10.2±6.2 |
| south and southeast Asia | 21.8 | 9.6 | 46.6 | 37.2±14.4 | 33.6±13.3 | 38.5±14.8 | 27.8±8.5 | 24.1±7.9 | 27.9±8.3 | 32.9±10.1 | 29.0±9.6 | 33.6±10.4 | 15.1±9.5 | 13.3±8.9 | 15.5±9.6 |
| Pacific developed region | 3.6 | -6.0 | 18.6 | 6.0±4.4 | 5.4±4.2 | 6.0±4.3 | 7.1±3.5 | 6.0±3.4 | 5.6±3.3 | 18.0±3.1 | 16.4±2.9 | 14.3±3.0 | -1.7±2.6 | -1.9±2.6 | -2.0±2.6 |
| north Africa and Middle East | 1.7 | -0.6 | 6.0 | 1.1±0.6 | 0.6±0.6 | 0.8±0.6 | 4.3±1.1 | 3.1±0.8 | 3.5±0.9 | 4.5±4.5 | 3.0±3.0 | 3.1±3.7 | -0.1±0.5 | -0.2±0.5 | -0.1±0.5 |
| globe | 148 | 94 | 273 | 152 ±49 | 154 ±50 | 159 ±51 | 161 ±40 | 162 ±39 | 163 ±39 | 165 ±46 | 160 ±45 | 161 ±47 | 119 ±37 | 121 ±38 | 122 ±38 |

