# Peer review of "Land-use and land-cover change carbon emissions between 1901 and 2012 constrained by biomass observations"

_Biogeosciences, 2017_

## Referee Comment (RC1) · Anonymous Referee #1 · 3 Jul 2017

Li et al add to the growing number of manuscripts on emergent constraints. Here they arrive at constrained LULCC emissions estimates by combining TRENDYv2 output, observationally-inspired biomass datasets, and regressions. This paper is well written. I have only a few very very minor issues (a 30 minute time burden at most to fix). Otherwise, publish as is.

P4L1: Help the reader who is not steeped in the minutiae of LULCC terminology here by adding more detail on "lost sink capacity" and "the loss of additional sink capacity"

P5L23: Incomplete sentence starting with Liu et al. (2015) ...?

P9L8: I appreciate the understatement but unless I misread Table 3 all the estimates

are (by row) indistinguishable in a statistical sense. So you might want to focus on that and relative error to paint a slightly more optimistic picture of robustness.

P10L13: Remove "the" before "CO2 fertilization"

P10L15: The "Methods of defining" sentence. Not sure how to read the "to be associated". There is no future tense here? Just rewrite this to highlight the robustness of your findings.

P10L27: Your "that" clause is off. Perhaps start a new sentence with "This may..."? You use 1/3 of mean biomass later on (same para) and use the word large here. Can you quantify large so we have some sense of scale regarding the 1/3 number?

P16: Not sure what to make of the nonsense words here...

My final point (take it or leave it, it's more of a meta-point, as it were) is more of a "so what" question. Looking at Figure 6 (and excluding the Pan et al. bars) what has the gain in all this been? To put it another way, the "best estimate" from TRENDYv2 would be the 150 value. That was your "new and improved" value? I am not trying to belittle this effort or mindset. This is simply a question I've had whenever I read an emergent constraint paper. I'm also not sure there are enough papers out to form a critical mass to inspire a "best practices" or "lessons learned" paper. But it's thought to keep in mind.

---

## Referee Comment (RC2) · Anonymous Referee #2 · 28 Aug 2017

This study's main aim is to reduce uncertainty on the magnitude of land use change emissions over the past century or so by constraining estimates from individual models with the use of remotely sensed biomass. The idea is that uncertainty in model-derived LUC emissions is partly due to uncertainty in the biomass state at the beginning of the model simulation period, which is partially fitting because much of the LUC emissions derives from the live biomass itself. While the basic concept is good, the study adopts an odd approach that seems poorly guided by logic. The study also fails to fully evidence uncertainty reduction. Furthermore, it misses an equally large if not larger concern about across-model spread in biomass and how that contributes to uncertainty in LUC emissions. These and other concerns are elaborated upon below. These

concerns notwithstanding, the study has significant merit overall, and involves several powerful new datasets on land use change and biomass, that if properly incorporated, could serve to significantly advance understanding of land change emissions.

1) A logic concern: The approach oddly ignores the vast discrepancy across models with regard to their estimates of biomass. One could directly use remotely sensed biomass in the 2000s to more directly evaluate which models match the data. Instead the authors do some contortions: (1) convert present-day observed biomass to year 1901 biomass based on year-2000 biomass versus year-1901 biomass, wherein even the use of a regression seems incorrect, and then (2) applies the expected year 1901 biomass according to observations in a regression equation of across-model LUC emissions versus each model's initial biomass estimate to derive a sort of model-guided distribution of inferred LUC emissions. This is a strange approach. In a sense, it estimates the LUC emissions we would expect, on average according to models, given an initial year-1901 biomass that has been estimated from contemporary observations and then adjusted to remove intervening effects of growth and mortality. It is rather indirect. It does not reduce across-model spread in mean emissions estimates, nor does it evaluate models against biomass observations. The method should really be improved with something more direct.

2) A logic concern: The approach seems to suggest that we should trust each model's relation between LUC emissions and biomass, and that all of the spread or uncertainty is due solely to model error in biomass. However, there are several other causes of across model spread in LUC emissions estimates and these get ignored in the present approach.

3) A logic concern: Regarding the alleged reduction in cumulative LULCC emissions, it seems to be the trivial and obvious result of replacing the across-model spread with results from a regression of across-model results. Of course this reduces uncertainty, but it needs to do so based on additional information or understanding and the only additional information comes from an odd and flawed use of biomass observations as

a sort of indirect constraint.

4) A technical concern: Regarding the use of regression to estimate biomass in 1901 consistent with year 2000 observations of biomass, regression seems to be the wrong tool here. Instead of an across-model regression, it would seem more appropriate to use an across-model mean delta biomass (2000 minus 1901 biomass change). The regression equation has an intercept that does not have a proper meaning in this case.

5) An interpretation concern: The across method spread (method A, B and C) in the constrained emissions estimates is still rather large but this is largely ignored in the presentation with a preference to show uncertainty reduction relative to the initial across-model spread in singular estimates. This leads to the next point.

6) An interpretation concern: The new uncertainty is inadequately compared to the original across-model spread in cumulative LULCC emissions. Here are some concerns and suggestions for how this could be corrected. (a) Figure 6: Use boxplots for all bars, or bars for the TRENDY estimates. Mixing these two display approaches in the figure adds confusion and obfuscates the main comparison of interest. Also make the use of error bars or percentile whisker's consistent across the TRENDY to constrained estimates. These changes will allow readers to clearly see the degree to which uncertainty reduction has been achieved with the constraint. Right now we can't see that. (b) Figure 3: Add the probability density function of the unconstrained TRENDY model cumulative LULCC emissions to each panel's right-side graph. This could have a light shaded red for each model's distribution, and a dark, thick red line for a Bayesian combination across the models. (c) Table 3 and Figure 5: Mixing an uncertainty spread shown as min/max, or range, for the TRENDY results compared to a 1 stdev spread for this study's inferred estimates inhibits a clear comparison of uncertainties across these two approaches. This needs to be corrected.

7) Model output data concern: TRENDY V2 LUC simulations had a problem with the fluxes for the final decade simulated. This study probably needs to switch to use of V4,

or maybe to shorten the study to exclude the period that was in error. Please check with your co-author Stephen Sitch with any questions.

8) Additional details:

Clarify in the methods how effects of LUC on biomass up to the time of observation (year 2000s) were accounted for in the inference of year 1901 biomass. The methods were not clear on this point.

Figure 4 should probably be presented before Figure 3 based on the methodological flow for the study.

Figure 4: explain in the figure caption that both biomass terms are from TRENDY, not the remote sensing

Remove the junk text after the citations (1.2 Subsection, 1.2.1 Subsection)...

---

## Author Comment (AC1) · 6 Oct 2017

**Response to comments**

**Paper #:** *bg-2017-186*
**Title:** *Land-use and land-cover change carbon emissions between 1901 and 2012 constrained by biomass observations*
**Journal:** *Biogeosciences*

**Reviewer #1:**

**General Comments:**

**Comment:**

Li et al add to the growing number of manuscripts on emergent constraints. Here they arrive at constrained LULCC emissions estimates by combining TRENDYv2 output, observationally-inspired biomass datasets, and regressions. This paper is well written. I have only a few very very minor issues (a 30 minute time burden at most to fix). Otherwise, publish as is.

**Response:**

We thank the reviewer for the comments and suggestions. Please see the detailed point-by-point responses below.

**Specific Comments:**

**Comment:**

P4L1: Help the reader who is not steeped in the minutiae of LULCC terminology here by adding more detail on "lost sink capacity" and "the loss of additional sink capacity"

**Response:**

The sentence on **P4L1** was revised as "This calculation of LULCC emissions by DGVMs includes the "lost sink capacity" (called "altered sink capacity" in Gasser and Ciais, (2013) and "loss of additional sink capacity" in Pongratz et al., (2014)), because simulated NBP in the S2 simulation without LULCC is a net sink over areas affected by LULCC in S3. For example, forests have larger carbon storage and slower turnover time than croplands and thus are expected to be carbon sinks when atmospheric $CO_2$ level increases. After deforestation to croplands, this sink capacity due to $CO_2$ fertilization is lost."

**Comment:**

P5L23: Incomplete sentence starting with Liu et al. (2015) ...?

**Response:**

This sentence was revised as: "Note that the uncertainties in the corresponding constrained results using these three alternative datasets do not include 1) the uncertainties of converting aboveground biomass to the total of aboveground and belowground biomass for datasets from Liu et al. (2015) and GEOCARBON (Avitabile et al., 2016; Santoro et al., 2015) and 2) the uncertainties of ignoring non-woody biomass in datasets from GEOCARBON (Avitabile et al., 2016; Santoro et al., 2015) and Pan et al. (2011)."

**Comment:**

P9L8: I appreciate the understatement but unless I misread Table 3 all the estimates are (by row) indistinguishable in a statistical sense. So you might want to focus on that and relative error to paint a slightly more optimistic picture of robustness.

**Response:**

As suggested, we added a sentence on **P9L8**: "The estimates of $E_{LUC}^c$ constrained by the biomass datasets from Liu et al. (2015) and GEOCARBON (Avitabile et al., 2016; Santoro et al., 2015) are rather consistent with $E_{LUC}^c$ constrained by biomass data from Carvalhais et al. (2014), implying the robustness of our estimates."

We didn't expand this point because the three alternative observation-based biomass datasets from Liu et al. (*2015*), GEOCARBON (*Avitabile et al., 2016; Santoro et al., 2015*) and Pan et al. (*2011*) are not fully consistent (in terms of the inclusion of herbaceous biomass, the underlying forest cover map, the measurement date, ...) (see **Section 2.2**) and are not entirely independent from the biomass dataset of Carvalhais et al. (*2014*) (e.g. In the NH, the basis of Carvalhais et al. (*2014*) should be very similar to Santoro et al. (*2015*). And Carvalhais et al. (*2014*) and Avitabile et al. (*2016*) uses Saatchi et al. (*2011*) for the tropics).

**Comment:**

P10L13: Remove "the" before "CO2 fertilization"

**Response:**

Revised accordingly.

**Comment:**

P10L15: The "Methods of defining" sentence. Not sure how to read the "to be associated". There is no future tense here? Just rewrite this to highlight the robustness of your findings.

**Response:**

This sentence was revised as: "For a given biomass dataset, the choice of a method for defining deforestation grid cells (Method-A, Method-B and Method-C) has very small influence on our results (Table 3)."

**Comment:**

P10L27: Your "that" clause is off. Perhaps start a new sentence with "This may..."? You use 1/3 of mean biomass later on (same para) and use the word large here. Can you quantify large so we have some sense of scale regarding the 1/3 number?

**Response:**

This sentence was revised as: "Currently, the uncertainties in the satellite-based biomass datasets are relatively large (e.g. 38% on average in tropics, Saatchi et al., 2011) at the pixel level (< 1 km). This introduces uncertainties in the constrained cumulative LULCC emissions, depending on the forest types and biomass range. For example, on average at the global scale, the relative uncertainty at the resolution of DGVM grid cells ($0.5° \times 0.5°$) is about one-third of the mean biomass (Carvalhais et al., 2014) and the relative uncertainty is smaller for high biomass areas in tropics (Avitabile et al., 2016; Saatchi et al., 2011)."

**Comment:**

P16: Not sure what to make of the nonsense words here...

**Response:**

We are sorry about this editing error, and this page was deleted accordingly.

**Comment:**

My final point (take it or leave it, it's more of a meta-point, as it were) is more of a "so what" question. Looking at Figure 6 (and excluding the Pan et al. bars) what has the gain in all this been? To put it another way, the "best estimate" from TRENDYv2 would be the 150 value. That was your "new and improved" value? I am not trying to belittle this effort or mindset. This is simply a question I've had whenever I read an emergent constraint paper. I'm also not sure there are enough papers out to form a critical mass to inspire a "best practices" or "lessons learned" paper. But it's thought to keep in mind.

**Response:**

We thank the reviewer for this thought. We acknowledge that the constrained cumulative LULCC emissions are close to the unconstrained ones from models. This is what the results turned out to be. Our work offers at least an evaluation of the modeling results using the observation-based biomass. More importantly, we combined the uncertainties in the regressions from state-of-the-art models with uncertainties in multiple observation-based biomass and gave a constraint with a 1-σ Gaussian

uncertainty. The idea of an emergent constraint paper is to give a more accurate estimate and / or a reduced uncertainty on an unknown variable by combining a heuristic relationship between two modeled variables (an observable and an unknown one) with actual observations of the observable variable. "Lessons learned" in our study are 1) that there is a heuristic relationship between biomass and cumulative land use emissions among different models, 2) that available biomass data confirm independently the median of modeled emission estimates, and 3) that more accurate biomass data in the future would allow to falsify some of the modelled estimates of emissions. We will elaborate on these points at the end of the conclusion section in the revised manuscript.

---

## Author Comment (AC2) · 6 Oct 2017

**Response to comments**

**Paper #:** *bg-2017-186*
**Title:** *Land-use and land-cover change carbon emissions between 1901 and 2012 constrained by biomass observations*
**Journal:** *Biogeosciences*

**Reviewer #2:**

**General Comments:**

**Comment:**

This study's main aim is to reduce uncertainty on the magnitude of land use change emissions over the past century or so by constraining estimates from individual models with the use of remotely sensed biomass. The idea is that uncertainty in model-derived LUC emissions is partly due to uncertainty in the biomass state at the beginning of the model simulation period, which is partially fitting because much of the LUC emissions derives from the live biomass itself. While the basic concept is good, the study adopts an odd approach that seems poorly guided by logic. The study also fails to fully evidence uncertainty reduction. Furthermore, it misses an equally large if not larger concern about across-model spread in biomass and how that contributes to uncertainty in LUC emissions. These and other concerns are elaborated upon below. These concerns notwithstanding, the study has significant merit overall, and involves several powerful new datasets on land use change and biomass, that if properly incorporated, could serve to significantly advance understanding of land change emissions.

**Response:**

We thank the reviewer for the comments and suggestions. Please see the detailed point-by-point responses below. We clarified the logic concerns and added two supplementary approaches to estimate the constrained cumulative LULCC emissions ($E_{LUC}^c$).

For the across-model spread in biomass, in fact we did include it in our approach using biomass observation to constrain LUC emissions from the initial range of models. The principle of the emergent constraint is to use the spread in biomass and in $E_{LUC}^c$ across models to build a heuristic relationship between both variables, so that observation of biomass can be used to "verify" and / or constrain $E_{LUC}^c$. Thus, the across-model spread in biomass is reflected in the regressions found between $E_{LUC}^c$ and biomass, and we included the regression error in the constrained $E_{LUC}^c$ as well as the errors on observed biomass (**Eq 1** on **P6L31**).

**Comment:**

1) A logic concern: The approach oddly ignores the vast discrepancy across models with regard to their estimates of biomass. One could directly use remotely sensed biomass in the 2000s to more directly evaluate which models match the data. Instead the authors do some contortions: (1) convert present-day observed biomass to year 1901 biomass based on year-2000 biomass versus year-1901 biomass, wherein even the use of a regression seems incorrect, and then (2) applies the expected year 1901 biomass according to observations in a regression equation of across-model LUC emissions versus each model's initial biomass estimate to derive a sort of model- guided distribution of inferred LUC emissions. This is a strange approach. In a sense, it estimates the LUC emissions we would expect, on average according to models, given an initial year-1901 biomass that has been estimated from contemporary observations and then adjusted to remove intervening effects of growth and mortality. It is rather indirect. It does not reduce across-model spread in mean emissions estimates, nor does it evaluate models against biomass observations. The method should really be improved with something more direct.

**Response:**

As we described above, our approach **doesn't** ignore the discrepancy in the simulated biomass across models. In fact, a fundamental basis of the emergent constraint method is that model biases in biomass are related to biases in $E_{LUC}^c$ so that observations of biomass can be used to provide a

constrained estimate of a target variable (similar constraint papers like *Cox et al., 2013; Wenzel et al., 2016; Kwiatkowski et al., 2017*).

We agree that the observation-based biomass in the 2000s could be directly used to evaluate the models performance on biomass estimates. This work has already been done (for northern boreal and temperate forests) by *Thurner et al.* (*2017*, see reference below) using the ISI-MIP Fast Track simulations and with a focus on carbon turnover processes. The objective of our study, however, is to constrain the modelled $E_{LUC}^c$ from biomass observation rather than to explore the differences in present-day biomass between models. Because land use emissions are related to the biomass that have been affected since the start of the land use perturbation, only biomass in 1901 (rather than that left out of land use in the 2000s) in LULCC-affected grid cells is **logically related** to historical $E_{LUC}^c$. Thus, converting present-day biomass to biomass in 1901 is a more direct and process-justified approach compared to regressing present-day biomass versus $E_{LUC}^c$ which has less logical basis.

As described on **P10L10**, we acknowledge that the regression to extrapolate the biomass in 1901 is a statistical model and cannot separate the effects of different contributing factors on the biomass changes (e.g. LULCC itself, as well as the intervening effects of growth and mortality mentioned by the reviewer). However, this biomass regression is applied only for the LULCC-affected grid cells, in order to reconstruct biomass in 1901 that has been exposed to LULCC during the 20th Century. The initial biomass in 1901 is larger than the present-day biomass (see **Figure 4**, all the markers are above the 1:1 line). Therefore, we would expect that LULCC is the main driving factor for the biomass change, because other dominating factors such as $CO_2$ fertilization would drive the biomass change to the reverse direction (i.e., a larger present-day biomass than 1901). In addition, the biomass regression is statistically significant for all regions and has a very high value of $r^2$ (>0.92 in seven regions, **Figure 4**), which gives confidence to our biomass extrapolation.

Still, we offer the reviewer to test two supplementary methods to constrain $E_{LUC}^c$: 1) regression between $E_{LUC}^c$ and present-day biomass, despite that this approach is not justified by a logical mechanism; and 2) using ΔB (biomass difference between present-day biomass and biomass in 1901 derived from the model simulations) instead of a regression between biomass in 1901 and present-day biomass to extrapolate the observation-based biomass in 1901.

The global constrained $E_{LUC}^c$ using these two supplementary methods is almost identical with that using our original method. The difference at the global level is **<1%** for all biomass observation datasets (Carvalhais et al. (*2014*), Liu et al. (*2015*), GEOCARBON (*Avitabile et al., 2016; Santoro et al., 2015*) and Pan et al. (*2011*)) and all methods to select LULCC grid cells (Method-A, -B and -C). This suggests that our constrained results are very robust. The change in the uncertainty in global constrained $E_{LUC}^c$ is also very small (**<2%**), because most of the uncertainties are from the observations and the regression between $E_{LUC}^c$ and biomass, rather than converting present-day biomass to biomass in 1901. One example is given in the figure below. The difference in regional $E_{LUC}^c$ between different constraint methods is relatively larger (12% on average), but the difference remains very small in tropical regions (~1%).

We will describe these two supplementary methods and their results in the revised manuscript.

**Figure R1** The global constrained $E_{LUC}^c$ using the original method (white bars), regression between $E_{LUC}^c$ and present-day biomass (light gray bars) and ΔB (dark gray bars). In this example the observation-based biomass dataset from Carvalhais et al. (*2014*) is used.

[Figure]

**Reference:**

*Thurner, M., Beer, C., Ciais, P., Friend, A. D., Ito, A., Kleidon, A., Lomas, M. R., Quegan, S., Rademacher, T. T., Schaphoff, S., Tum, M., Wiltshire, A. and Carvalhais, N.: Evaluation of climate-related carbon turnover processes in global vegetation models for boreal and temperate forests, Global Chang. Biol., 23(8), 3076–3091, doi:10.1111/gcb.13660, 2017.*

*Cox, P. M., Pearson, D., Booth, B. B., Friedlingstein, P., Huntingford, C., Jones, C. D. and Luke, C. M.: Sensitivity of tropical carbon to climate change constrained by carbon dioxide variability, Nature, 494(7437), 341–344, doi:http://www.nature.com/nature/journal/v494/n7437/abs/nature11882.html#supplementary-information, 2013.*

*Kwiatkowski, L., Bopp, L., Aumont, O., Ciais, P., Cox, P. M., Laufkötter, C., Li, Y. and Séférian, R.: Emergent constraints on projections of declining primary production in the tropical oceans, Nat. Clim. Chang., 7(5), 355–358, doi:10.1038/nclimate3265, 2017.*

*Wenzel, S., Cox, P. M., Eyring, V. and Friedlingstein, P.: Projected land photosynthesis constrained by changes in the seasonal cycle of atmospheric CO2, Nature, 538(7626), 499–501, doi:10.1038/nature19772, 2016.*

**Comment:**

2) A logic concern: The approach seems to suggest that we should trust each model's relation between LUC emissions and biomass, and that all of the spread or uncertainty is due solely to model error in biomass. However, there are several other causes of across model spread in LUC emissions estimates and these get ignored in the present approach.

**Response:**

We agree that the LULCC emissions are not solely affected by biomass. Other factors like crop / wood harvest practice and soil carbon change also contribute to the LULCC emissions.

However, we do not think that it is a logic problem that we established an empirically true relationship between LULCC emissions and biomass across models, which allows us to apply an emergent constraint. First, the essence of the emergent constraint approach is to find a heuristic relationship and then to combine observations to reduce the uncertainty. It is not necessary to fully explain it or include all factors while using this method (e.g. *Cox et al., 2013; Wenzel et al., 2016; Kwiatkowski et al., 2017*). In our case, we offered an explanation on **P3L7** and **P4L9**, that biomass in 1901 subjected to LULUC during the 20[th] Century represents the dominant carbon stocks involved in LUC, and thus explains most of the cumulative land use emissions. Soil carbon stocks affected by land use, in contrast, are much smaller, in particular in tropical forests where most of the land use change takes place today. We will further emphasize this point in the revised manuscript. The empirical regression between modelled $E_{LUC}^c$ and biomass in 1901, which serves as the basis of the emerging constraint, is statistically significant and substantial (e.g. $r^2 = 0.66$ at global scale, **Figure 3**) and thus supports to constrain $E_{LUC}^c$ through biomass observations. Second, the relationship between $E_{LUC}^c$ and biomass reflects the current scientific understanding that biomass loss is the largest contributor to $E_{LUC}^c$ (e.g. a net carbon loss of 110 Pg C from biomass due to land use change during 1850-1990, accounting for 89% of total $E_{LUC}^c$ of 124 Pg C, according to *Houghton, 1999*).

In addition, although soil carbon changes after land use change are another important component in the LULUC flux, soil carbon largely comes from biomass, e.g. dead roots after forest clearing. The

lack of global observation-based soil carbon dataset also hinders the application of emergent constraint method including soil carbon stock as an additional variable.

We added the following sentences to clarify these points on **P10L17**: "LULCC carbon emissions are influenced not only by changes in biomass, but also by how these are prescribed in the model to influence posterior changes in detrital and soil organic carbon pools. However, LULCC emissions are dominated by changes in biomass. For example, LULCC results in a net carbon loss of 110 Pg C in biomass during 1850-1990, accounting for 89% of the total $E_{LUC}^c$ (Houghton, 1999). The soil carbon changes after LULCC is also indirectly impacted by initial biomass, since the dead roots and remaining aboveground debris turn into soil organic carbon after land clearing, which takes longer time to return into the atmosphere. In addition, it is not necessary to account for all factors when applying an emergent constraint approach (e.g. Cox et al., 2013; Wenzel et al., 2016; Kwiatkowski et al., 2017). The regression between $E_{LUC}^c$ and biomass in 1901 in models in our study is satisfying (e.g. $r^2 = 0.66$ at global scale, Figure 3) to constrain $E_{LUC}^c$ through biomass observations."

**Reference:**

Cox, P. M., Pearson, D., Booth, B. B., Friedlingstein, P., Huntingford, C., Jones, C. D. and Luke, C. M.: Sensitivity of tropical carbon to climate change constrained by carbon dioxide variability, Nature, 494(7437), 341–344, doi:http://www.nature.com/nature/journal/v494/n7437/abs/nature11882.html#supplementary-information, 2013.

Houghton, R. A.: The annual net flux of carbon to the atmosphere from changes in land use 1850-1990, Tellus B, 51(2), 298–313, doi:10.1034/j.1600-0889.1999.00013.x, 1999.

Kwiatkowski, L., Bopp, L., Aumont, O., Ciais, P., Cox, P. M., Laufkötter, C., Li, Y. and Séférian, R.: Emergent constraints on projections of declining primary production in the tropical oceans, Nat. Clim. Chang., 7(5), 355–358, doi:10.1038/nclimate3265, 2017.

Wenzel, S., Cox, P. M., Eyring, V. and Friedlingstein, P.: Projected land photosynthesis constrained by changes in the seasonal cycle of atmospheric CO2, Nature, 538(7626), 499–501, doi:10.1038/nature19772, 2016.

**Comment:**

3) A logic concern: Regarding the alleged reduction in cumulative LULCC emissions, it seems to be the trivial and obvious result of replacing the across-model spread with results from a regression of across-model results. Of course this reduces uncertainty, but it needs to do so based on additional information or understanding and the only additional information comes from an odd and flawed use of biomass observations as a sort of indirect constraint.

**Response:**

Our objective is not to understand why models have different biomass. The emergent constraint approach doesn't necessarily require a perfect mechanism explanation either. As described above, our method of using biomass to constrain $E_{LUC}^c$ is direct and logic. The "additional information", i.e. the biomass datasets, represents the most advanced biomass observations, and the "understanding", i.e. the regressions from state-of-the-art models, is also robust (see $r^2$ of the regressions in **Figure 3** and **Figure 4**) and makes a lot of sense (see explanations above).

**Comment:**

4) A technical concern: Regarding the use of regression to estimate biomass in 1901 consistent with year 2000 observations of biomass, regression seems to be the wrong tool here. Instead of an across-model regression, it would seem more appropriate to use an across-model mean delta biomass (2000 minus 1901 biomass change). The regression equation has an intercept that does not have a proper meaning in this case.

**Response:**

As suggested, we added supplementary analyses using mean ΔB between 1901 and present day across models. The results show no difference from our original estimates (see the response and **Figure R1** above).

However, we should note that ΔB is not a perfect solution to extrapolate biomass in 1901 from present-day biomass, because the change in biomass is not solely impacted by land use change. The interactions between biomass and climate conditions, disturbances and nutrient limitation are also very important in DGVMs. For example, historical land use change (LUC) may reduce biomass over LUC-affected regions, e.g. by replacing forests with croplands. On the contrary, the $CO_2$ fertilization effects may increase biomass over LUC and non-LUC regions. Therefore, ΔB reflects a mixed effect of different factors, not a sole response to LUC. In addition, as the ΔB has a higher relative uncertainty between models (~53% at the global level), using the regression ($r^2 > 0.92$ in seven regions, **Figure 4**) to calculate biomass in 1901 could include relatively less noisy information than using ΔB.

**Comment:**

5) An interpretation concern: The across method spread (method A, B and C) in the constrained emissions estimates is still rather large but this is largely ignored in the presentation with a preference to show uncertainty reduction relative to the initial across-model spread in singular estimates. This leads to the next point.

**Response:**

As suggested, we gave an ensemble mean value of Method-A, -B and –C and included the spread of results from these three methods in the uncertainty of ensemble mean. We added an ensemble mean bar for each observation-based biomass dataset in **Figure 6** (reproduced below).

**Comment:**

6) An interpretation concern: The new uncertainty is inadequately compared to the original across-model spread in cumulative LULCC emissions. Here are some concerns and suggestions for how this could be corrected. (a) Figure 6: Use boxplots for all bars, or bars for the TRENDY estimates. Mixing these two display approaches in the figure adds confusion and obfuscates the main comparison of interest. Also make the use of error bars or percentile whisker's consistent across the TRENDY to constrained estimates. These changes will allow readers to clearly see the degree to which uncertainty reduction has been achieved with the constraint. Right now we can't see that. (b) Figure 3: Add the probability density function of the unconstrained TRENDY model cumulative LULCC emissions to each panel's right-side graph. This could have a light shaded red for each model's distribution, and a dark, thick red line for a Bayesian combination across the models. (c) Table 3 and Figure 5: Mixing an uncertainty spread shown as min/max, or range, for the TRENDY results compared to a 1 stdev spread for this study's inferred estimates inhibits a clear comparison of uncertainties across these two approaches. This needs to be corrected.

**Response:**

We thank the reviewer for these suggestions. The reason that we used mixed ranges to display the original TRENDY estimates and our constrained results is that we don't know the probability distribution of the original TRENDY estimates whereas the 1-σ Gaussian uncertainty of our constrained results come from the biomass observation errors (assumed Gaussian unbiased) and the regression errors. Thus to be accurate, we gave the quartile ranges of original TRENDY estimates rather than using the standard deviations across models in the original manuscript version (The estimates from all individual models can be seen from the y-axis of **Figure 3**).

If we assumed that the original TRENDY estimates followed a normal distribution, the mean and standard deviation (158 ± 48 Pg C) are close to the constrained estimate and its 1-σ Gaussian uncertainty (155 ± 50 Pg C). We would like to note that it doesn't mean that our emerging constraint method is not effective, but that the relatively large uncertainty of the constrained $E_{LUC}^c$ is propagated from the biomass observation uncertainty, which is about one-third of the mean biomass (see **P10L29**). We also emphasize that there is no reason to believe that the original TRENDY estimates is normally distributed across models, whereas our methods provide a biomass-constrained estimate with its 1-σ Gaussian uncertainty. Secondly, model estimates are just unconstrained. Here we independently verified the median model estimate by biomass observation, giving independent

support to the use of this estimate in carbon budget assessments (and falsifying some individual DGVM models as well).

As suggested, we added the means and standard deviations of original TRENDY estimates in **Figure 5**, **Figure 6** and **Table 3** on the top of the quartile ranges. We will also add sentences to discuss the probability distribution of the original TRENDY estimates in the revised manuscript.

For (b), we cannot give a distribution for each model because each model only simulated one $E_{LUC}^c$ value, and it is difficult to give a Bayesian combination across models without knowing the distribution form of the error of each model, and of the cross models results.

**Figure R2** (**Figure 6**) The global cumulative land-use and land-cover change (LULCC) emissions during 1901-2012 from original TRENDY models and from the estimates constrained by different biomass datasets with different methods to define deforestation grid cells. "All methods" represents the ensemble mean and uncertainty of the constrained results from "Method-A", "Method-B" and "Method-C". The whisker-box plot represents the minimum and maximum values, 25th and 75th percentiles and the median value of original TRENDY models. The light red dot and error bars show the mean and standard deviation of the original TRENDY estimates. The error bars in the bar plot represent the 1-σ Gaussian errors.

[Figure]

**Comment:**

7) Model output data concern: TRENDY V2 LUC simulations had a problem with the fluxes for the final decade simulated. This study probably needs to switch to use of V4, or maybe to shorten the study to exclude the period that was in error. Please check with your co-author Stephen Sitch with any questions.

**Response:**

We confirmed this issue with Stephen Sitch. The problem with the LUC fluxes for the final decade is caused by the difference in gross land area changes (i.e. forest becoming crops and crops becoming forests in the same time in a region) during the **1990s** between HYDE3.1 used as the LUC area forcing data in TRENDY v2 (this study) and HYDE3.2 used for TRENDY v4. The global gross area changes during 1990s were larger in HYDE3.1 than in HYDE3.2, which means more positive and negative changes using HYDE3.1, but the global net changes are rather consistent.

In this study, however, we used the **cumulative** LUC emissions during 1901-2012. We compared the difference in LUC fluxes between TRENDY v2 and v4 models. The difference in cumulative LUC emissions during 1990-2012 (the period that may be influenced by forcing data difference) is only **3%** on average of the cumulative emissions during 1901-2012 across models. Note that the difference may not be solely caused by difference in LUC forcing data and may also be caused by model developments like implementing new processes between model versions. In addition, as discussed on **P4L10**, although models shared the same LUC forcing data, they have different PFT types and different rules of translating the LUC forcing data into changes in PFTs, which may result in larger difference in LUC emissions than the difference in the original LUC forcing data.

Therefore, the influence of the difference between HYDE3.1 and HYDE3.2 during 1990s on our results is negligible. We keep using TRENDY v2 outputs in our study.

**Specific Comments:**

**Comment:**

8) Additional details:

Clarify in the methods how effects of LUC on biomass up to the time of observation (year 2000s) were accounted for in the inference of year 1901 biomass. The methods were not clear on this point.

**Response:**

As suggested, we added sentences on **P5L2** in **Methods**: "LULCC can either reduce or increase the biomass amount over time depending the LULCC types. For example, forest clearing turns the forest biomass into atmospheric $CO_2$ eventually, while secondary forest regrowth can increase biomass. The overall effects of LULCC on biomass during the historical period is a net loss of carbon (Houghton, 1999) due to converting natural vegetation into cultivated lands by human (Klein Goldewijk et al., 2011)."

**Comment:**

Figure 4 should probably be presented before Figure 3 based on the methodological flow for the study.

**Response:**

We agree that based on the methodological flow, **Figure 4** is before **Figure 3**. However, we mentioned the original $E_{LUC}^{c}$ from models (**Figure 3**) in Section 3.1, thus earlier than the biomass relationship (**Figure 4**). So we keep **Figure 3** before **Figure 4**.

**Comment:**

Figure 4: explain in the figure caption that both biomass terms are from TRENDY, not the remote sensing.

**Response:**

Revised accordingly.

**Comment:**

Remove the junk text after the citations (1.2 Subsection, 1.2.1 Subsection)...

**Response:**

Deleted accordingly.

---

## Author Response (AR1)

**Response to comments**

**Paper #:** *bg-2017-186*
**Title:** *Land-use and land-cover change carbon emissions between 1901 and 2012 constrained by biomass observations*
**Journal:** *Biogeosciences*

**Editor:**

**General Comments:**

**Comment:**

Response comments and proposed edits are satisfactory overall, and if implemented fully I expect the manuscript will be accepted for publication. There are, however, some remaining issues I would ask the authors to address.

**Response:**

We thank the editor for the suggestions. We fully implemented the proposed edits in the revised manuscript. Please see the point-by-point responses below.

**Comment:**

The abstract ends with the claim that constraint from biomass observations reduced the uncertainty of the historical carbon budget, however this does not appear to be consistent with the results of the manuscript. The study reports that the mean and standard deviation on the constrained LUC emissions is almost identical to that from the TRENDY DGVMs if one assumes that they have a normal distribution. It would thus be difficult to argue that uncertainty has been reduced and such statements need to be removed from the paper. You could potentially claim to have identified one or more of the TRENDY members that are inconsistent with the observations. You can also comment, as in the Conclusions, that the wide range in the constrained estimates derives from wide uncertainty in biomass observations which acts as the constraint, and that reductions in uncertainties on observed biomass would improve the constraint that these observations could provide.

**Response:**

We went through the manuscript and revised the related sentences. All the "reduced uncertainty" description has been deleted. We pointed out that the wide range in the constrained estimates are from the biomass observations in Abstract and also in Results and Discussion.

**Comment:**

R1 asks a good question about what this study offers given the above result. The so-called "lessons learned" are well communicated in the response comment and should be reflected in the paper's discussion and/or conclusions.

**Response:**

As suggested, we added these "lessons learned" in Conclusion.

**Comment:**

Given concerns about non-normality of the distribution of ELUC estimates across the TRENDY DGVM ensemble, you might consider using quantiles for both the original _and_ the constrained estimates when you compare them.

**Response:**

As suggested, we added the interquantiles of global constrained $E_{LUC}^c$ in **Figure 6** and gave the interquantile ranges for each region in **Table S1**. Texts are also revised accordingly.

[revised manuscript text omitted]
 | 148 | 94 | 273 | 152±49 | 154±50 | 159±51 | 161±40 | 162±39 | 163±39 | 165±46 | 160±45 | 161±47 | 119±37 | 121±38 | 122±38 |